# Mechanically Robust and Flexible GO/PI Hybrid Aerogels as Highly Efficient Oil Absorbents

**DOI:** 10.3390/polym14224903

**Published:** 2022-11-13

**Authors:** Li Zhang, Yuting Wang, Ruidong Wang, Penggang Yin, Juntao Wu

**Affiliations:** Key Laboratory of Bio-Inspired Smart Interfacial Science and Technology of Ministry of Education, School of Chemistry, Beihang University, Beijing 100191, China

**Keywords:** GIAs, compressive strength, absorption capacity

## Abstract

Herein, mechanically robust and flexible graphene oxide/polyimide (GO/PI) hybrid aerogels (GIAs) were fabricated by a facile method, in which the mixed suspensions of the water-soluble polyimide precursor and graphene oxide (GO) sheets were freeze-dried, which was followed by a routine thermal imidation process. The porous GIAs obtained not only exhibit excellent elasticity and extremely low density values (from 33.3 to 38.9 mg.cm^−3^), but they also possess a superior compressive strength (121.7 KPa). The GIAs could support a weight of up to 31,250 times of its own weight, and such a weight-carrying capacity is much higher than that of other typical carbon-based aerogels. Having such a porous structure, and high strength and toughness properties make GIAs ideal candidates for oil spill cleanup materials. The oil/organic solvents’ absorption capacity ranges from 14.6 to 85, which is higher than that of most other aerogels (sponges). With their broad temperature tolerance and acidic stability, the unique multifunctional GIAs are expected to further extend their application range into extreme environments.

## 1. Introduction

With increasing industrial oily wastewater generation and numbers of oil spill accidents, materials that can selectively absorb oil from wastewater are becoming greatly desired [1,2,3,4,5]. The interconnected porous structures of aerogels could offer a passageway to absorb, store, remove and transport oil and organic liquids [6,7,8,9]. As a result, various inorganic aerogels, including SiO_2_ aerogels, carbon nanotube sponges and graphene aerogels have been made to remove oil from wastewater [10,11,12]. However, inorganic aerogels generally suffer from weakened mechanical properties, extreme brittleness and a loss of functionality. Traditional polymer aerogels, on the other hand, are flexible, but they are relatively weak in extreme environments, such as in high and low temperatures and acidic conditions [13,14,15,16]. To meet the specific requirements of such applications, the integration of nanofillers and polymers to form hybrid aerogels absorbents is one of the most promising strategies for taking full advantage of their unique structures and properties. As a result, several methods have been reported to take advantage of this combination. The first method involves nanocasting conformal inorganic coatings on the preformed 3D porous skeletons of aerogels [17,18], however, the inorganic coating can be easily detached when the sorbent is manually squeezed [19]. The infiltration method sacrifices the high inner surface area of the aerogel, as the open pores are filled with inorganics [20]. Alternatively, mixing a suspension of the polymers or their precursors with inorganic materials was found to be an efficient method to make hybrid aerogels with considerable mechanical properties [20,21,22,23,24]. Considering their practical application, a more facile approach and more robust and flexible absorbent materials are essential as well.

As a type of high-performance polymeric material, polyimide (PI) features superior mechanical strength and flexibility properties, a high glass transition temperature, as well as excellent thermal and chemical stability properties [25]. These features make PI an attractive candidate for aerogel with high oils and organic solvents absorbency, selectivity, and excellent recyclability. In our previous works, we fabricated PI aerogels with efficient oil/water separation and reusability even in extreme environments, e.g., high or low temperatures and harsh acid environments [26,27]. However, the hydrophobic and hydrophilic groups endow the PI with amphiphilic characteristic [28]. Instead, it should be more hydrophobic for aerogels to possess high absorption capacity and oil/water selectivity for practical applications. Furthermore, the previously reported PI aerogels have ordinary mechanical properties, which cannot satisfy the versatile applications in extreme environments, especially those with large mechanical stresses or thermal shocks.

To solve these problems and further enhance the mechanical strength of the PI aerogels, graphene oxide (GO) was chosen as the nanofiller due to its water solubility, prominent mechanical strength and flexibility, large surface-to-volume ratios, and it having plenty of oxygen-containing groups on its surface [29], which enable the formation of strong hydrogen bonds with PI and its precursor. Freeze-drying was adopted to prepare hybrid aerogels as a cost-effective and environmentally friendly approach that can shape advanced materials in various geometries [30]. In this study, highly robust and flexible GO/PI hybrid aerogels (GIAs) were designed and facilely fabricated, which involved mixing the water-soluble PI precursors and GO sheets and freeze-drying and thermal imidizing the resulting suspension. The GIAs can efficiently, in a selectively and recyclable way, absorb various oils and organic solvents with high absorbencies of up to 85 times their own weight.

## 2. Materials and Methods

### 2.1. Materials

4,4′-Oxydianiline (4,4′-ODA, 99.8%, CAS: 101-80-4) and pyromellitic dianhydride (PMDA, 99.5%, CAS: 89-32-7) were purchased from Beijing Chemical Reagents Company (Beijing, China). Triethylamine (TEA, 99.0%, CAS: 121-44-8) and N,N-dimethylacetamide (DMAc, 99.5%, CAS: 127-19-5) were purchased from Beijing Chemical Works (Beijing, China). Graphite oxide (CAS: 149-91-7) was purchased from Nanjing Xianfeng Nanomaterials Technology Co., Ltd. (Nanjing, China). Deionized water (CAS: 7732-18-5) was used in the experiments. All of the chemical reagents were used without further purification.

### 2.2. Preparation of GIAs via the Freeze-Drying Method

The GIAs were prepared according to Figure 1. Into a 100 mL three-necked round-bottom flask, 0.012 mol 4,4′-ODA (2.4 g) was added, and it was mechanically mixed with 36.784 g DMAc at room temperature. After the ODA was completely dissolved, equivalent moles of PMDA (2.616 g) were added slowly to the above solution, and the mixture was stirred for 4 h at room temperature. After a continuous stirring for another 4 h, a poly(amic acid) (PAA) solution with a solid content of 12 wt% was obtained, which was poured into deionized water, and it was allowed to deposit afterwards. PAA powder was obtained by washing, drying, and crushing the precipitate. After adding PAA (1 g) and triethylamine (TEA, 0.48 g) into varying amounts of deionized water, such as 0, 27.62, 27.11, and 26.08 g of it, and continuously stirring until the solutions became homogeneous, the ammonium salt solutions of PAA (PAS) were prepared. GO suspension with a mass fraction of 1 wt% was prepared by exfoliating 0.1 g graphite oxide in 9.9 g deionized water with the aid of sonication for 30 min. Typically, the as-prepared GO suspension with varying amounts, such as 0, 0.5025, 1.0101, and 2.0408 g of it, were added to the previously prepared PAS solutions, correspondingly. Afterwards, the resultant solutions were continuously stirred until they were homogeneous. Later, the obtained solutions were poured into to 5 mL cylindrical vials, frozen at −18 °C in a laboratory freezer, and freeze-dried for 24–36 h in a lyophilizer. Finally, the resulting samples were taken out and thermally imidized in an oven at 100 °C for 1 h, at 120 °C for 1 h, at 150 °C for 1 h, at 180 °C for 0.5 h, at 250 °C for 1 h and at 300 °C for 1 h step by step. A series of GO/PI hybrid aerogels with different GO contents were prepared and named as GIA_x_, where x is the weight ratio of GO to PAA and GO. As an example, after PAA (1 g) and TEA (0.48 g) in 27.62 g deionized water became homogeneous (the molar ratio of TEA and carboxyl groups on PAA molecular chains is 1:1), 0.5025 g GO suspension was poured into the vessel. The mixture was vigorously mixed using an electric mixer for hours to obtain the GO/PAS solution with a solid content of 5 wt% and to create the GIA_0.5_ sample.

### 2.3. Characterizations and Instruments

The microstructures of GIAs were observed by SEM (Quanta 250 FEG). The wetting properties of the GIAs were analyzed through contact angle tests (DCAT21 contact angle analysis system) at room temperature. The densities of the GIAs were calculated from the ratio of mass/volume. The bulk masses were measured using an electronic balance. The dimensions and heights were measured using a slide caliper to calculate the bulk volumes. The compression tests of the GIAs were performed using a testing machine (Instron 5843) with a 1000-N load cell, and the crosshead speed was maintained at 10 mm/min.

### 2.4. Organic Liquids Absorption of GIAs

The absorption capacities (Q) of the GIAs were measured by immersing them into and removing them from various organic solvents and oils with different surface tensions, including ethanol, toluene, cyclohexane, Arawana cooking oil and glycerol. The weight measurements of the wet GIAs should be performed quickly to avoid the evaporation of absorbed organic liquids. Q was calculated according to the following equation:Q = S_t_/S_0_(1)
where S_0_ is the weight of dry GIA, and S_t_ is the total weight of wet GIA.

## 3. Results and Discussion

### 3.1. Morphology of GIAs

According to Figure 1, hybrid aerogels were designed and facilely fabricated, in which the mixed suspensions of the PAS and GO sheets were freeze-dried and thermally imidized. As shown in Figure 1a,b, GIA_0_, i.e., the pure PI aerogel, changed from being white to brown in appearance, and the higher content of the GO was, the darker the hybrid aerogels were. Moreover, due to the extreme lightness that originated from their porous structure, the GIAs could rest atop stably on a bristlegrass, without causing any deformation to the bristlegrass (Figure 1c,d). As demonstrated in Figure 1e–h, the GIAs possessed 3D porous interconnected networks with finely dispersed GO in a PI matrix, and most of the pores were continuously distributed at several micrometers to dozens of micrometers apart. Unlike the dense pores observed in GIA_0_, these larger and looser pores with similar honeycomb-like structures, which may be caused by freezing-induced orientation [31], were observed in GIA_0.5_, GIA_1_ and GIA_2_. The densities of the hybrid GIAs (GIA_0.5_, GIA_1_ and GIA_2_) ranged from 0.033 to 0.039 g·cm^−3^ (Appendix A), and they showed linear relationship with respect to the GO contents. This relationship indicates that the porous structures of the GIAs are similar to each other [32], which is consistent with the observed microstructure in the SEM images (Figure 1e–h). It is noteworthy to state that the shrinkage of GIA_0_ has led to a much higher density than that of the hybrid GIAs, and the addition of GO can reduce the drying shrinkage and the aerogel density [33]. The explanation of these finding lies in the formation mechanisms of the GIAs. On the one hand, during the freeze-drying process, the pore size difference created a high capillary pressure gradient which further led to the shrinkage or even collapse of the porous structure [34]. On the other hand, the capillary tension during drying was too large [34], and so multiple non-covalent bonds of PAS chains [35] were not strong enough to withstand the capillary pressure, resulting in the shrinkage of the GIAs. Compared with the pristine ones, the GIAs with GO shrunk much less, and the reasons for this are likely twofold (Appendix A). Firstly, because of its excellent mechanical strength and strong covalent interaction with PAS, GO can significantly fortify the structures and inhibit the shrinkage of cryogels during the drying process. Secondly, GO can act as a huge barrier between the PAS chains and obviously hinder the connection of the chains, which may reduce the shrinkage or even collapse of the cryogels as well [33]. The higher the GO loading is, the heavier the weights of the GIAs are, and with similar porous structures, the larger the densities of the GIAs are.

### 3.2. Mechanical Properties of GIAs

Besides acting as an anti-shrinkage additive to lessen the structural shrinkage, GO can also play another important role in enhancing the compressive strength and toughness of the GIAs. During the loading process, the stress–strain curves exhibited two distinct stages (Figure 2a–d). The linear-elastic regime, at ε < 15%, which is demonstrated by the elastic bending of cell walls, and the non-linear regime, at 15% < ε < 50%, which features an increased slope where the deformation is still recoverable because of the elastic buckling of the cell walls. Such behaviors are consistent with previous reports [36,37,38]. Similar to most resilient cellular materials [11,31,39], the GIAs also suffered from hysteresis. As shown in Figure 2, hysteresis loops were found in the loading−unloading cycles, indicating energy dissipation that can be ascribed to the sliding of graphene and the rupture of the sacrificial bonds. Detailed discussions on this topic are given later. As shown in Figure 2e, the maximum compressive strengths of GIA_0.5_ and GIA_1_ at the 50% cyclic compression strain showed 27.4% and 28.6% increases over that of GIA_0_, respectively. However, as the GO content increased from 1 to 2 wt%, the maximum compressive strength of the GIAs decreased from 121.7 to 112.4 KPa, probably because the GO content exceeded the critical level and formed small agglomerates [40]. At low percentage content of GO, the special combination of covalent and non-covalent interactions between the PAS chain and the GO nanosheets helped to improve the mechanical properties of the hybrid aerogels, and this involved three possible interactions (Appendix A) [41,42,43]: (1) H-bonding between the O-atoms of the PAS chains and the COOH-functional group of the GO nanosheets; (2) π–π interaction between the PAS chain and the GO nanosheets; (3) the carboxyl groups on the graphene surfaces or edges may react with the amido groups of PAS during the high-temperature imidization process [41]. Surprisingly, the maximum compressive strength of the various GO-containing GIAs shown in this work is higher than that of many other aerogels (sponges) which were reported at a 50% strain (Figure 2e) [11,20,44,45,46,47,48,49,50,51]. Moreover, GIA_1_ with mass of 6.4 mg can support a counterweight of 200 g, which is at least 31250 times of its own weight, without a sign of deformation (Appendix A). Judging from this, our hybrid aerogels are much more robust than most aerogels that have been reported previously are [52,53,54,55,56,57,58,59,60,61,62]. The weight-carrying capacity of the GIAs in this work is approximately thirty times higher than that of carbon nanotube aerogels [63] and about six times higher than that of a graphene sponge [64] and a reduced graphene oxide-konjac glucomannan carbon aerogel [58]. As far as we know, the weight-carrying capacity of our aerogels is only lower than that of rGO aerogels, which can support about 100,000 times their own weight [65] and supercritically dried PI aerogel which is able to support the weight of a car [66]. However, these two types of areogels encounter severe difficulty during industrialization and in meeting the requirements of possible applications. On one hand, it is difficult for the rGO aerogels be used in industry applications due to their relatively poor compressive strength and toughness. On the other hand, for the supercritically dried PI aerogels, five factors can be taken into consideration. (1) Industrially, curing is carried out by the direct heating of the PI precursor at high temperatures, instead of chemical imidization which is adopted for the supercritically dried PI aerogel, to ensure the complete imidization and conversion of undesirable isoimides to imides [67]. (2) Compared with supercritical drying, the freeze-drying method that is adopted in our paper is more facile, cost-effective and environmentally friendly. (3) As a strong covalently bonded network structure may sacrifice its toughness [68,69], it cannot be applied in pressure sensors as they cannot be squeezed for oil spill recyclability. (4) With primary micropores and mesopores, it is difficult for the oils to permeate into the supercritically dried PI aerogels [70], which may hamper their utilization as oil sorbents. (5) From the standpoint of their application, the high density (ranging from 131 to 333 mg·cm^−3^) of the supercritically dried PI aerogels are not the preferred choice for lightweight devices. As such, with marvelous strength and toughness properties, our hybrid aerogels would be ideal candidates for various applications [71].

The intriguing high elasticity and mechanical robustness of our hybrid aerogels may be attributed to the low densities and unique honeycomb-like microstructures of them, the outstanding mechanic properties of PI and GO, as well as the synergistically covalent and non-covalent interactions between the PI chain and the GO nanosheets. As shown in Figure 3a, under compression, the motion of the PI chains during deformation caused the cracking of the π-π stacking and therefore, the energy that dissipated in this process resulted in an increased toughness of the GO-free aerogel. When the GO-free aerogel was severely compressed, the reconstructed π-π stacking between the PI chains allowed for the van der Waals adhesion to be overcome by the elastic energy and the GO-free aerogel spring back to its original volume afterward. GO can be partially reduced during the thermal imidization process [41]. The functional groups that remained may react with the PI chains and the π-π stacking interaction, hydrogen bonding and covalent bonding formed between the PI chains and partially reduced GO, and graphene may also form π-π stacking interaction with the PI chains and the partially reduced GO. Upon compression, the motion of the PI chains during deformation may engender the sliding of the graphene and the breaking of π-π bonds (sacrificial bonds), and thus, this provides an energy dissipation approach (Figure 3b). The dissociation of the hydrogen bonds (sacrificial bonds) resulting from further increasing of the load may enable additional energy dissipation. With an enormous surface area, the partially reduced GO and graphene can endow numerous sites for the formation of sacrificial bonds, i.e., hydrogen bonds and π-π bonds. All of these can ensure the elasticity of the hybrid aerogels which are comparable to the GO-free aerogel. The mechanical robustness of hybrid aerogels is ascribed to the densely clustering of multiple non-covalent interactions of the sacrificial bonds and the strong interaction between the carboxyl groups of the partially reduced GO and the amido of PAA.

### 3.3. Oil Absorption Properties of GIAs

It is well known that surface wettability is determined by the geometric structure and the surface energy [72,73,74]. On one hand, with porous microstructures and a low surface energy, the PI aerogels can be highly hydrophobic [26]. On the other hand, during the curing process, GO undergoes thermal reduction, and the hydrophilic functional groups on the surface of GO are partially reacted, thus is turns into a more hydrophobic partially reduced GO and graphene. Therefore, it is possible to achieve more hydrophobic hybrid aerogels after the addition of graphene with a low surface energy. As expected, more hydrophobic hybrid aerogels are prepared (Appendix A). Interestingly, the water contact angles of the hybrid aerogels increase with an increasing GO content, which is in consistent with previous reports [75,76].

Inspired by the porous framework, the superior mechanical properties, high hydrophobicity and high absorption capacities of PI aerogels at high/low temperature and in harsh acid environments were demonstrated in our previous works [26,27], and we also consider hybrid aerogels to be an ideal candidate for highly efficient sorbents. Various kinds of organic solvents and oils, e.g., Arawana cooking oil (Figure 4a), were investigated, and the hybrid aerogels exhibit very high absorption capacities for all of them. In general, the hybrid aerogels could absorb the liquids from 14.6 to 85 times their own weights (Figure 4b), which is in the range of values that have been reported in the literature for similar materials [10,17,21,77,78,79,80,81,82,83,84,85,86,87]. The absorption capacities of the GIAs increased with increasing GO loading as well, and three reasons may be ascribed to this phenomenon. (1) Compared to GO-free aerogels, the hybrid aerogels with less shrinkage have a larger pore size and a larger pore volume to store more absorbed liquid [70]. (2) During the curing of hybrid aerogels, the GO surface can decompose into CO_2_ and water, and thus, voids and interconnected tortuous pathways can be generated, which further increase the surface area and pore volume. (3) Sheet-like graphene provide a tremendous surface area, which facilitates the adsorption of oil [88]. The absorption capacity is determined by the surface tension of liquid. The following formula can be used to calculate the mass (m) of liquid penetrating into the pores of aerogel [89]:2πrγcosθ = mg(2)

When the surface of the liquid is completely wetted, the contact angle θ is zero, and for such surfaces:2πrγ = mg(3)
or
m = kγ(4)
where γ is the surface tension of the liquid, r is the pore radius of the aerogel, g is the gravitational acceleration, and k = 2πr/g. Therefore, according to Equation (4), the absorption capacities of the GIAs increase linearly with an increase in the surface tension of the liquid (the surface tensions of these organic liquids [89,90,91] are listed in Appendix A), as demonstrated in Figure 4c.

## 4. Conclusions

To prepare aerogels with high absorption capacity and mechanical strength properties, GIAs were produced by freeze-drying of water-soluble PI precursor/GO suspensions, which was followed by a multistep thermal imidation. The resulting GIAs exhibited an extremely low density (33.3–38.9 mg.cm^−3^), a high absorption capacity (14.6–85) and superior compressive strength (121.7 KPa). Moreover, the GIAs could support a weight of up to 31,250 times their own weight. This research provides a versatile platform for fabricating hybrid aerogels with synergistically ameliorated multifold properties, such as reduced shrinkage and high hydrophobicity properties and a high absorption capacity. Furthermore, the honeycomb-like structures and the dense clustering of multiple non-covalent interactions of the sacrificial bonds and their strong interaction give rise to excellent flexibility as well as considerable compressive strength of hybrid aerogels. The successful synthesis of such fascinating GIAs with good mechanical properties, broad temperature tolerance, acidic resistance, high hydrophobicity and a high absorption capacity paves the way to explore aerogels for multifunctional practical applications, especially in extreme environments.

## Data Availability

The data presented in this study are available on request from the corresponding author.

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
