# Peer review of "Mechanically Robust and Flexible GO/PI Hybrid Aerogels as Highly Efficient Oil Absorbents"

_polymers, 2022, doi:10.3390/polym14224903_

Round 1

Reviewer 1 Report

The present work is interesting, well-desinged, well-performed and clearly discussion. 

Some minor remarks

i. The use of sample codes in abstract should be avoided.

ii. The use of English language should be checked, especially in cases of long sentences.  

Reviewer 2 Report

For Mechanically Robust and Flexible GO/PI Hybrid Aerogels as Highly Efficient Oil Absorbents, the paper could be accepted after minor revision. The CAS numbers for all used chemicals should be presented in the Experimental part of the paper. The english language should be improved Accept after minor revision (corrections to minor methodological errors and text editing)

Reviewer 3 Report

In this paper, the authors document and discuss the excellent compressive mechanical properties of GO/PI hybrid aerogels and their strong adsorption to organic solvents. However, certain concerns need to be addressed:

1.In the preparation process, the specific ratio of various reagents can be written more specifically.

2.In section 3.3, the mentioned CO2 does not appear above, please explain how CO2 is produced and decomposed.

3.The conclusion is too short and must be improved, highlighting the purpose of the paper, summarizing the methods used and the results achieved.

4.There are some grammar and word mistakes in the manuscript. Please go through the manuscript carefully again.
